# Beyond Two-Stage Training: Cooperative SFT and RL for LLM Reasoning

Liang Chen [1]   Xueting Han [2]   Li Shen [3]   Jing Bai [2]   Kam-Fai Wong [1]

## Abstract

Supervised fine-tuning (SFT) and reinforcement learning with verifiable rewards (RLVR) are two widely used post-training paradigms for improving the reasoning ability of large language models (LLMs). Recent methods attempt to integrate SFT and RLVR in a single stage by reweighting or scheduling their objectives. However, such coupling can be counterproductive because supervised updates are not uniformly beneficial for reward optimization. To address this, we propose BRIDGE, a scalable framework in which SFT learns to supervise RL by selectively transferring knowledge that improves reward optimization. Specifically, BRIDGE alternates two updates at each meta-training step: a base-model update that fuses the SFT and RL gradients, and an update to a lightweight low-rank adapter (LoRA) that coordinates the two objectives by maximizing a cooperative-gain signal, defined as the reward of joint SFT–RL training over an RL-only baseline. Across five mathematical reasoning benchmarks, BRIDGE consistently outperforms two-stage cold start, naive mixing, and representative single-stage integration baselines, yielding over three points average absolute improvement and more stable training dynamics. We further show that BRIDGE extends to logical reasoning and generalizes out-of-distribution to code and science without additional training, while staying robust under noisy rewards.

## 1. Introduction

Large reasoning models (LRMs) have demonstrated strong performance across a range of domains, particularly in challenging tasks such as mathematics (Cobbe et al., 2021; Hendrycks et al., 2021) and programming (Chen et al., 2021; Codeforces, 2025). Two post-training paradigms are widely used to elicit such reasoning capabilities: supervised fine-tuning (SFT) (Muennighoff et al., 2025) and reinforcement learning with verifiable rewards (RLVR) (DeepSeek-AI et al., 2025). These paradigms offer complementary strengths. SFT can mimic high-quality expert trajectories efficiently, but it is prone to overfitting (Chu et al., 2025). RLVR, in contrast, encourages the policy to actively explore reward-yielding trajectories, which can improve generalization (Song, 2025; Jiang et al., 2023), but it is inefficient due to trial-and-error search. A common recipe therefore uses a two-stage SFT-then-RL pipeline. However, this pipeline does not consistently outperform pure RL (Table 2), as also reported in prior work (Zhang et al., 2025b;a). These observations motivate more effective approaches to integrating the two paradigms.

Existing methods for integrating SFT and RLVR for reasoning can be grouped into two categories. First, *objective-level combination* integrates SFT and RL by weighting or scheduling their objectives, ranging from interleaved recipes (e.g., alternating RL and SFT when RL stalls) (Ma et al., 2025) to single-stage multi-objective training with adaptive reweighting or gating (Zhang et al., 2025b; Chen et al., 2025a; Fu et al., 2025). Second, *data-augmented RL* incorporates SFT data as off-policy trajectories within the RL objective (Yan et al., 2025), typically weighted by importance-sampling ratios to mitigate distribution mismatch; however, it often underperforms objective-level combination in practice (Zhang et al., 2025b; Chen et al., 2025a). Despite their practical success, objective-level combinations rarely characterize how the two learning signals interact. As shown in Figure 2, we find that a simple combination of SFT and RL updates even decrease the reward of RL, indicating that not all supervised updates are helpful for reward optimization.

In this work, we address the challenges by formulating the integration as a meta-learning problem, BRIDGE, which treats SFT as an upper-level *teacher* and RL as a lower-level *student*. By modeling the hierarchical structure—with the SFT objective explicitly conditioned on the RL solution—we enable SFT to provide *targeted* guidance that directly supports RL optimization, rather than updating SFT and RL independently and balancing them via heuristics.

[1]The Chinese University of Hong Kong [2]Microsoft Research Asia [3]Shenzhen Campus of Sun Yat-sen University. Correspondence to: Xueting Han <chrihan@microsoft.com>, Kam-Fai Wong <kfwong@se.cuhk.edu.hk>.

*Proceedings of the $43^{rd}$ International Conference on Machine Learning*, Seoul, South Korea. PMLR 306, 2026. Copyright 2026 by the author(s).

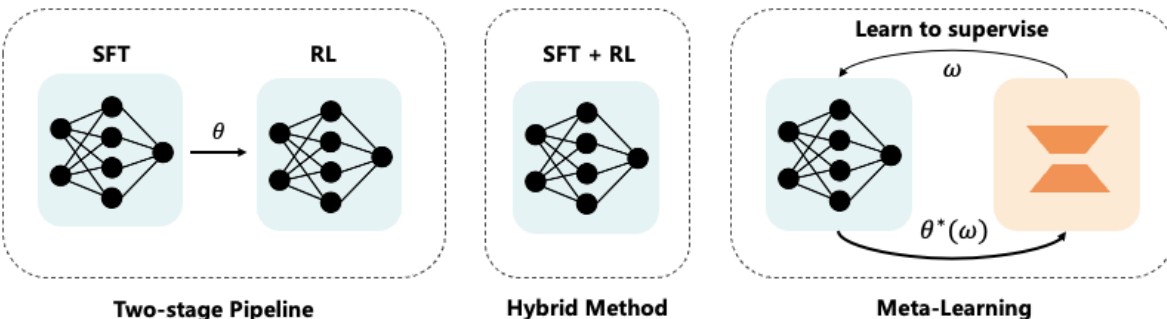

*Figure 1.* Comparison of three training paradigms. **Left:** The two-stage pipeline first performs SFT, then RL, with unidirectional knowledge transfer. **Middle:** The single-stage hybrid training, which combines SFT and RL objectives via weighting or scheduling on the same parameters without modeling their interaction. **Right:** Our meta-learning approach introduces a teacher module ($w$) that learns to supervise the student LLM ($\theta^*(w)$), enabling bidirectional adaptation between the two objectives.

A direct bilevel solver typically requires second-order derivatives (Finn et al., 2017; Hu et al., 2023), which are prohibitive at LLM scale. To reduce computational overhead, we adopt a first-order, penalty-based relaxation. Concretely, the lower level updates mix two objectives, while the upper level updates maximizes the *cooperative gain*—the reward difference of mix SFT-RL training over RL training alone. We further separate these roles across parameter types: the lower level updates the LLM parameters (the student), whereas the upper level updates a newly initialized low-rank adapter (LoRA; Hu et al., 2021) as *meta-parameters* (the teacher). This design yields an efficient, scalable algorithm suitable for large-scale training.

To validate the effectiveness of our approach, we conduct comprehensive experiments with three LLMs of varying scales on five diverse math reasoning benchmarks. Results demonstrate that BRIDGE consistently outperforms all baselines, including SFT, RL, two-stage pipelines, and recent hybrid training methods. Notably, BRIDGE requires less wall-clock training time than the two-stage method while delivering superior performance, highlighting its practical efficiency. Furthermore, extensive ablation studies confirm the necessity of the bilevel cooperation design and demonstrate the robustness of our method to hyperparameter variations across different model sizes and task difficulties. Beyond in-domain mathematics, BRIDGE also generalizes to out-of-distribution reasoning—logical puzzles, code generation, and scientific QA—without any additional training, preserves solution diversity (consistently higher Pass@$K$ than RL), and degrades gracefully under noisy rewards, indicating that its benefits are not tied to mathematics or to perfectly clean verifiers. These improvements confirm the benefits of coupling SFT and RL through bilevel optimization, enabling the model to selectively learn from supervised signals that contribute to reward maximization. The code will be available at https://github.com/ChanLiang/BRIDGE.

**Conflict of Interest Disclosure.** The authors declare no financial conflicts of interest related to this work.

## 2. Background and Preliminaries

We review two prevalent post-training paradigms for reasoning in LLMs—supervised fine-tuning (SFT) and reinforcement learning with verifiable rewards (RLVR)—and discuss a widely used hybrid objective. We show that combining the two objectives does not necessarily improve reward and can sometimes lead to lower reward.

### 2.1. Fine-tuning Methods for Reasoning Models

Let $\pi_\theta(y \mid x)$ denote a language model with parameters $\theta$ that defines a conditional distribution over output sequences $y$ given an input prompt $x$. We assume a reasoning dataset $\mathcal{D} = \{(x, r, y)\}$, where $x$ is an input question, $y$ is a verifiable target answer, and $r$ is an expert reasoning trace. During training, SFT and RLVR operate on different views of the same dataset: SFT uses $(x, r, y)$, while RLVR uses $(x, y)$ to compute rewards.

**SFT.** Given a question $x$, SFT maximizes the log-likelihood of the expert trace $r$ and final answer $y$ jointly:

$$J_{\text{SFT}}(\theta) := \mathbb{E}_{(x,y,r)\sim\mathcal{D}} \big[ \log \pi_\theta(r, y \mid x) \big]. \quad (1)$$

This objective encourages the LLM to imitate expert reasoning patterns and to produce the corresponding answer.

**RLVR.** RLVR does not require annotated reasoning traces. Given a question $x$, the policy samples a reasoning trace $\hat{r}$ and an answer $\hat{y}$, and receives a reward based on answer correctness. A standard KL-regularized objective is:

$$\begin{aligned} J_{\text{RL}}(\theta) := \mathbb{E}_{(x,y)\sim\mathcal{D},\ (\hat{r},\hat{y})\sim\pi_\theta(\cdot|x)} \big[ R(\hat{y}, y) \big] \\ - \beta_{\text{KL}} \, \mathbb{E}_{x\sim\mathcal{D}} \Big[ D_{\text{KL}} \big( \pi_\theta(\cdot \mid x) \,\|\, \pi_{\text{ref}}(\cdot \mid x) \big) \Big]. \end{aligned} \quad (2)$$

where $\pi_{\text{ref}}$ is a fixed reference policy and $\beta_{\text{KL}} \geq 0$ controls the strength of KL regularization. Here $R(\hat{y}, y)$ is computed by a deterministic, rule-based verifier (e.g., code execution or regular-expression matching). In practice, $J_{\text{RL}}$ is optimized using policy-gradient variants such as GRPO (DeepSeek-AI et al., 2025) and DAPO (Yu et al., 2025).

**Two-stage pipeline.** The prevailing paradigm (DeepSeek-AI et al., 2025) adopts a sequential protocol. The model is first optimized via $J_{\text{SFT}}$ to acquire foundational reasoning patterns (SFT warm-up), thereby providing a high-quality initialization for the subsequent maximization of $J_{\text{RL}}$. This approach mitigates the exploration difficulties inherent in training from scratch but decouples the supervised signal from the reward optimization phase.

**Single-stage hybrid methods.** Current methods often integrate SFT and RLVR by optimizing a weighted sum:

$$J_{\text{hyb}}(\theta) := J_{\text{RL}}(\theta) + \mu\, J_{\text{SFT}}(\theta), \qquad \mu \geq 0, \quad (3)$$

where $\mu$ trades off reward maximization and imitation learning. In practice, $\mu$ is set heuristically (e.g., fixed values, decay schedules (Zhang et al., 2025b), or adaptive rules based on entropy or gradient statistics (Fu et al., 2025)).

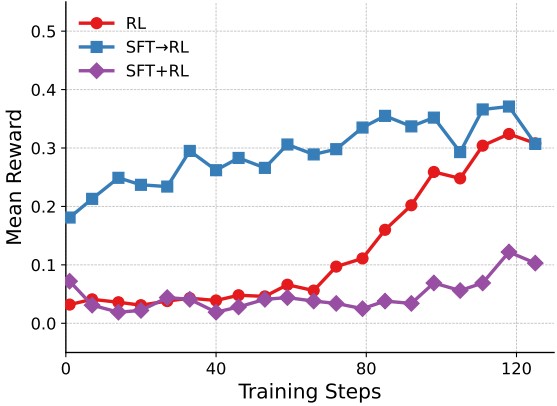

*Figure 2.* Reward comparison of training methods.

### 2.2. Analysis of Fine-Tuning Methods

To understand the interaction between supervised and reinforcement learning signals, we conduct a preliminary evaluation of three canonical fine-tuning paradigms on mathematics problems (Grades 3–5). We compare **RL** (training from scratch), **Two-Stage (SFT→RL)** (SFT followed by RL), and **Hybrid (SFT + RL)** (multi-task learning with a fixed scalar weight $\lambda = 1$). Figure 2 illustrates the mean reward trajectories throughout training.

The results highlight limitations in current methodologies. While **RL** (Red) eventually improves, it suffers from training inefficiency due to the lack of prior knowledge, requiring extensive exploration to discover high-reward regions.

Conversely, the **SFT→RL** approach (Blue) leverages SFT for a strong initialization. However, this advantage is primarily static. By discarding the supervised signal after the warm-up, the subsequent training phase reverts to unguided exploration. Consequently, the benefit of the SFT initialization is most pronounced in the early stages but diminishes as the model struggles to navigate the complex reasoning landscape without ongoing directional guidance. Notably, the naive **SFT+RL** method (Purple) yields the worst performance, significantly lagging behind even RL.

This suggests that not all supervised updates are beneficial for reward maximization; thus, we ask a natural question: *How can we dynamically extract the useful components of the supervised signal that actively facilitate the optimization of the RL reward?*

## 3. Methodology

We propose a meta-learning method that models the teacher-student relationship between SFT and RL. We first introduce the formulation, then present the learning algorithm.

### 3.1. BRIDGE: Meta-Learning for SFT and RL

Given a reasoning dataset $\mathcal{D}$ and an LLM parameterized by $\theta$ (defined in Section 2.1), our objective is to integrate the SFT objective (Eq.(1)) with the RL objective (Eq.(2)) such that SFT updates facilitate reward optimization in RL. We treat SFT as the *teacher*, since it has access to expert reasoning traces, and RL as the *student*, since it relies on policy exploration to discover high-reward traces. We model their relationship through the following bi-level optimization:

$$\begin{aligned} \max_{w} \quad & J_{\text{SFT}}(w, \theta^*(w)), \\ \text{s.t.} \quad & \theta^*(w) := \arg\max_{\theta} J_{\text{RL}}(\theta, w). \end{aligned} \quad (4)$$

where $\theta$ denotes the student LLM parameters and $w$ denotes the teacher's meta-parameters, instantiated as a lightweight LoRA module.

This formulation has a hierarchical structure inspired by Stackelberg games: SFT serves as the upper-level leader with access to RL's solution, providing supervision that improves reward optimization, while RL acts as the lower-level follower, optimizing the policy under guidance from SFT. This coupling enables the two objectives to cooperate dynamically, each adapting to the other's feedback.

Figure 1 contrasts our approach with prior paradigms. The two-stage pipeline and single-stage hybrid methods both apply two objectives to the same LLM either sequentially or simultaneously, without explicitly modeling their interaction. In contrast, our formulation introduces a separate teacher module $w$ (implemented as LoRA), which *learns to supervise* the student LLM. By conditioning each compo-

nent on the other's parameters, this design enables tighter coordination between the two learning signals.

**Why a LoRA teacher?** We use a lightweight LoRA teacher $w$ for three reasons: (i) since the policy is $\pi_{\theta+w}$, modifying $w$ reshapes the loss landscape that $\theta$ optimizes over, so the meta-objective steers *which* supervised gradients reach $\theta$ rather than rescaling a scalar weight; (ii) the separate low-rank subspace isolates the meta-update from $\theta$'s RL optimization; and (iii) once merged into the backbone, it adds zero inference-time cost.

### 3.2. Learning Algorithm via Penalty Relaxation

To efficiently solve the bi-level problem in Eq. (4) at LLM scale, we employ penalty-based methods (Shen & Chen, 2023; Shen et al., 2025) that avoid expensive second-order computations. We first reformulate Eq. (4) as a single-level problem, then apply first-order optimization.

We define the penalty function measuring the sub-optimality of the lower-level problem as:

$$p(w, \theta) = \max_{\theta'} J_{\mathrm{RL}}(\theta', w) - J_{\mathrm{RL}}(\theta, w). \quad (5)$$

which satisfies $p(w, \theta) = 0$ if and only if $\theta \in \arg\max_{\theta'} J_{\mathrm{RL}}(\theta', w)$. For $\lambda \in (0, 1)$, consider the following penalized objective:

$$\max_{\theta, w} \mathcal{L}(\theta, w) := (1 - \lambda) J_{\mathrm{SFT}}(\theta, w) - \lambda p(w, \theta). \quad (6)$$

**Intuition for the penalty relaxation.** The exact bilevel problem (Eq. (4)) requires $\theta$ to be RL-optimal, which is intractable at LLM scale. The reformulation in Eq. (6) softens this: instead of demanding that $\theta$ exactly maximize $J_{\mathrm{RL}}(\cdot, w)$, we penalize its sub-optimality $p(w, \theta)$. This is a natural penalty because (i) $p(w, \theta) = 0$ iff $\theta$ is RL-optimal, and (ii) whenever $p > 0$, $-\nabla_\theta p$ points toward RL optimality. Moreover, under standard smoothness assumptions the penalized solution $\theta_\lambda$ satisfies $\|\theta_\lambda - \theta^\star\| = O(1 - \lambda)$ relative to the exact bilevel optimum $\theta^\star$ (Shen & Chen, 2023), so a larger $\lambda$ yields a tighter approximation at the cost of weaker SFT influence.

**Update for $\theta$ (student).** Since $\max_{\theta'} J_{\mathrm{RL}}(\theta', w)$ does not depend on $\theta$, the update of LLM student is as follows:

$$\theta^{k+1} = \theta^k + \alpha \left[ (1 - \lambda) \nabla_\theta J_{\mathrm{SFT}}(\theta, w) + \lambda \nabla_\theta J_{\mathrm{RL}}(\theta, w) \right] \quad (7)$$

This yields a convex combination of SFT and RL gradients.

**Update for $w$ (teacher).** For the update of teacher's meta-parameters, we use Danskin's theorem. Assuming $J_{\mathrm{RL}}(\cdot, w)$ satisfies the required regularity conditions (e.g., $J_{\mathrm{RL}}(\theta, w)$

---

**Algorithm 1:** Learning Algorithm of BRIDGE

---
1: Initialize student parameters $\theta^0$, teacher parameters $w^0$, and auxiliary parameters $\hat{\theta}^0 := \theta^0$; dataset $\mathcal{D}$; learning rates $\alpha, \beta$; mixing coefficient $\lambda$; iterations $K$
2: **for** $k = 0$ to $K - 1$ **do**
3:     Sample mini-batches $\mathcal{B}_{\mathrm{SFT}} \sim \mathcal{D}$ and $\mathcal{B}_{\mathrm{RL}} \sim \mathcal{D}$
4:     // Compute base objectives
5:     Compute $J_{\mathrm{SFT}}(\theta^k, w^k)$ on $\mathcal{B}_{\mathrm{SFT}}$
6:     Compute $J_{\mathrm{RL}}(\theta^k, w^k)$ and $J_{\mathrm{RL}}(\hat{\theta}^k, w^k)$ on $\mathcal{B}_{\mathrm{RL}}$
7:     // Define composite objectives
8:     $J_{\mathrm{Joint}}(\theta^k, w^k) = (1 - \lambda) J_{\mathrm{SFT}}(\theta^k, w^k) + \lambda J_{\mathrm{RL}}(\theta^k, w^k)$
9:     Compute $J_{\mathrm{meta}}(\theta^k, w^k)$ according to Eq. (10)
10:    // Update student via joint objective (Eq. 7)
11:    $\theta^{k+1} \leftarrow \theta^k + \alpha \nabla_\theta J_{\mathrm{Joint}}(\theta^k, w^k)$
12:    // Update auxiliary parameters via RL objective
13:    $\hat{\theta}^{k+1} \leftarrow \hat{\theta}^k + \alpha \nabla_{\hat{\theta}} J_{\mathrm{RL}}(\hat{\theta}^k, w^k)$
14:    // Update teacher to maximize cooperative gain (Eq. 10)
15:    $w^{k+1} \leftarrow w^k + \beta \nabla_w J_{\mathrm{meta}}(\theta^k, w^k)$
16: **end for**

---

is differentiable in $w$ and $\arg\max_{\theta'} J_{\mathrm{RL}}(\theta', w)$ is non-empty), we have:

$$\nabla_w \max_{\theta'} J_{\mathrm{RL}}(\theta', w) = \nabla_w J_{\mathrm{RL}}(\theta^*(w), w), \quad (8)$$

where $\theta^*(w) = \arg\max_\theta J_{\mathrm{RL}}(\theta, w)$. In practice, we approximate $\theta^*(w)$ by taking a single gradient ascent step with respect to the RL objective: $\hat{\theta} = \theta + \alpha \nabla_\theta J_{\mathrm{RL}}(\theta, w)$, yielding the approximate gradient update for $w$:

$$\begin{aligned} \nabla_w \mathcal{L}_\lambda(\theta^k, w^k) = {} & (1 - \lambda) \nabla_w J_{\mathrm{SFT}}(\theta^k, w^k) \\ & + \lambda \left[ \nabla_w J_{\mathrm{RL}}(\theta, w) - \nabla_w J_{\mathrm{RL}}(\hat{\theta}, w) \right] \end{aligned} \quad (9)$$

To evaluate the cooperative-gain term in Eq. (9), at each iteration we draw a *single* batch of rollouts from $\pi_{\theta+w}$ and score both $\theta$ and $\hat{\theta}$ on these *shared* trajectories. Scoring on identical samples reduces variance and avoids sampling-noise confounding.

**Interpretation of teacher's behaviors** The update rule for the meta-parameter $w$ (Eq. (9)) can be interpreted as gradient ascent on the surrogate objective

$$\begin{aligned} J_{\mathrm{meta}}(\theta, w) = {} & (1 - \lambda) J_{\mathrm{SFT}}(\theta, w) \\ & + \lambda \left[ J_{\mathrm{RL}}(\theta, w) - J_{\mathrm{RL}}(\hat{\theta}, w) \right] \end{aligned} \quad (10)$$

where the first term preserves supervision from expert trajectories, and the second term is a *cooperative gain* signal measuring how much the joint SFT-RL model (using $\theta$) improves the RL objective relative to an RL-only one (using $\hat{\theta}$). Maximizing $J_{\mathrm{meta}}$ therefore encourages the teacher parameters $w$ to shape supervision based on its *utility for reward optimization*, rather than implicitly treating all supervised updates as uniformly beneficial.

Algorithm 1 presents the learning procedure for BRIDGE. At each iteration, we sample SFT and RL mini-batches;

update the base parameters $\theta$ with the joint objective; update the auxiliary parameters $\hat{\theta}$ via pure RL as a baseline; and optimize the LoRA parameters $w$ to maximize cooperative gain—the improvement of the joint objective over pure RL. This lets SFT meta-learn to guide RL's optimization.

## 4. Experiment

### 4.1. Settings

**Datasets.** We use the MATH dataset (Hendrycks et al., 2021) for RL training. Following the setup of SimpleRL (Zeng et al., 2025), we train on the *hard* split, which contains 8.5K problems with difficulty levels ranging from 3 to 5. For intermediate supervision, we use reasoning traces distilled from DeepSeek-R1 (DeepSeek-AI et al., 2025). For evaluation, we adopt five mathematical reasoning benchmarks: MATH500 (Hendrycks et al., 2021), Minerva Math (Lewkowycz et al., 2022), OlympiadBench (He et al., 2024), and two recent competition-level datasets—AMC 2023 and AIME 2024. To assess generalization beyond mathematics (Section 6), we additionally train on Knights-and-Knaves logic puzzles and evaluate zero-shot transfer of our math-trained checkpoints on two out-of-distribution benchmarks: LiveCodeBench (Jain et al., 2024) (code generation) and GPQA Diamond (Rein et al., 2024) (graduate-level science).

**Models.** To validate the generality of BRIDGE, we employ three LLMs spanning different families and scales: **Qwen2.5-3B** (Yang et al., 2024), a strong smaller-scale model from the Qwen2.5 family; **Llama-3.2-3B** (Grattafiori et al., 2024), an instruction-following model from Meta's Llama family, included to test cross-family generalization; and **Qwen3-8B-Base** (Yang et al., 2025), a larger model from the Qwen3 family used to assess performance at scale.

### 4.2. Baselines

We compare BRIDGE against a representative and comprehensive set of baselines, spanning widely used standard training recipes and major families of recent single-stage hybrid training methods. Specifically, we include: (i) **Original Model**: the base or instruction-tuned backbone without reasoning-specific training; (ii) **SFT**: imitation learning on expert reasoning traces; (iii) **RL** (Zeng et al., 2025): GRPO applied directly to the backbone without SFT warm-up; (iv) **SFT→RL (two-stage)** (DeepSeek-AI et al., 2025): the sequential pipeline that first performs SFT and then applies RL with decoupled objectives. We further compare against **single-stage hybrids** that integrate SFT and RL within a unified training stage: (v) **SFT+RL**: a multi-task baseline that directly combines and optimizes the SFT loss and the RL objective; (vi) **LUFFY** (Yan et al., 2025) (*data-augmented RL*): incorporates demonstration-style information into RL rollouts to stabilize and improve policy

optimization; (vii) **CHORD** (Zhang et al., 2025b): balances supervision and RL by dynamically weighting the objectives, including token-wise reweighting; (viii) **SRFT** (Fu et al., 2025): reduces interference between objectives via entropy-aware weighting and clipping. All baselines use the same backbones and training data.

### 4.3. Implementation Details

All models are trained with the VERL framework (Sheng et al., 2024) using GRPO, with a batch size of 128, mini-batch size of 64, learning rate $5 \times 10^{-7}$, 8 rollouts per prompt, and 2 epochs. The KL and entropy loss coefficients are $1 \times 10^{-4}$ and $1 \times 10^{-5}$, respectively. Maximum response length is 4K tokens for Qwen2.5-3B and 6K for both Llama-3.2-3B-Instruct and Qwen3-8B-Base. We evaluate with greedy decoding and report pass@1 accuracy. Experiments run on NVIDIA A100 and AMD MI300 GPUs. At inference, the teacher $w$ is merged into $\theta$ and we decode from a single model $\pi_{\theta+w}$, adding *zero* cost over the backbone.

### 4.4. Main Results

**Overall performance.** Tables 1–3 show that BRIDGE achieves the highest average accuracy across all three LLMs: 36.0% on Qwen2.5-3B, 24.7% on Llama-3.2-3B-Instruct, and 49.9% on Qwen3-8B-Base. These correspond to absolute gains of 3.5, 2.8, and 4.0 points over the respective strongest baselines. Notably, BRIDGE is the only method that consistently surpasses standard recipes (e.g., RL and the two-stage SFT→RL pipeline) across all settings, validating the efficacy of our meta formulation in extracting beneficial supervision without hindering reward optimization.

**Comparisons across baseline families.** Grouping baselines clarifies the role of different integration strategies. Among standard recipes, RL generally outperforms SFT on final accuracy, though less efficient. The two-stage pipeline (SFT→RL) proves a strong baseline, yielding consistent improvements on the Qwen family at both 3B and 8B.

For single-stage hybrids, naive objective-level mixing (SFT+RL) often produces performance between SFT and RL, indicating that simply summing objectives does not reliably improve upon RL—instead yielding a compromise solution. More sophisticated hybrids (e.g., CHORD, SRFT) employ heuristic weighting or scheduling of the two objectives, partially mitigating the adverse effects of supervision on RL exploration. However, these methods still leave a consistent margin to BRIDGE, which explicitly optimizes supervision to improve downstream reward gains rather than assuming that supervised updates are uniformly beneficial.

**Performance across model families and scales.** Baseline rankings vary across backbones, whereas BRIDGE remains consistently strong. For the strongest standard recipe, the

*Table 1.* Performance of BRIDGE compared to baselines on Qwen2.5-3B across five math benchmarks

| Method | MATH 500 | Minerva Math | Olympiad Bench | AIME24 | AMC23 | Average |
|---|---|---|---|---|---|---|
| Base | 32.4 | 11.8 | 7.9 | 0.0 | 20.0 | 14.4 |
| SFT | 53.4 | 18.8 | 21.5 | 3.3 | 42.5 | 27.9 |
| RL | 64.4 | 26.5 | 27.0 | 3.3 | 40.0 | 32.2 |
| SFT→RL | 66.0 | 24.3 | 26.8 | 9.0 | 35.0 | 32.2 |
| SFT+RL | 55.6 | 20.6 | 25.0 | 3.3 | 42.5 | 29.4 |
| LUFFY | 65.2 | 23.5 | 27.3 | 3.3 | 42.5 | 32.4 |
| SRFT | 62.6 | 22.1 | 24.4 | 9.0 | 37.5 | 31.1 |
| CHORD | 66.0 | 23.2 | 25.9 | 6.7 | 40.5 | 32.5 |
| BRIDGE | 66.2 | 23.9 | 28.9 | 13.3 | 47.5 | 36.0 |

*Table 2.* Performance on Llama3.2-3B-Instruct.

| Method | MATH 500 | Minerva Math | Olympiad Bench | AIME24 | AMC23 | Average |
|---|---|---|---|---|---|---|
| Instruct | 38.0 | 14.3 | 13.0 | 13.3 | 25.0 | 20.7 |
| SFT | 38.4 | 10.3 | 11.9 | 3.3 | 27.5 | 18.3 |
| RL | 48.6 | 15.1 | 17.8 | 10.0 | 17.5 | 21.8 |
| SFT→RL | 45.0 | 11.8 | 12.0 | 3.3 | 22.5 | 18.9 |
| SFT+RL | 45.8 | 13.6 | 17.3 | 3.3 | 20.0 | 20.0 |
| LUFFY | 49.0 | 14.0 | 17.1 | 6.7 | 22.5 | 21.9 |
| SRFT | 45.4 | 13.6 | 15.4 | 3.3 | 17.5 | 19.0 |
| CHORD | 46.0 | 14.3 | 17.9 | 3.3 | 22.5 | 20.8 |
| BRIDGE | 51.8 | 15.1 | 19.3 | 10.0 | 27.5 | 24.7 |

two-stage SFT→RL pipeline is competitive on Qwen 3B and 8B but underperforms RL on Llama-3B-Instruct, indicating that this fixed recipe, though strong, is not uniformly reliable across model families. Among hybrid baselines, CHORD is comparatively stronger on the Qwen backbones, while LUFFY is slightly stronger on Llama-3.2-3B-Instruct. Despite this variability, BRIDGE improves over the strongest baseline across different LLM families and preserves its advantage when scaling from 3B to 8B parameters. The gap between BRIDGE and the best hybrid widens from 3.5 points at 3B to 4.0 points at 8B, suggesting that principled SFT–RL coordination compounds with model capacity.

**Generalization to challenging benchmarks.** A critical limitation of SFT-based baselines is their tendency to plateau on challenging tasks. As shown in Table 3, while the two-stage pipeline (SFT→RL) improves performance on the standard benchmarks like MATH500, it underperforms RL in generalization on the more challenging OlympiadBench and Minerva Math. This indicates that supervised warm-up can restrict the policy to local optima. In contrast, BRIDGE preserves RL's superior generalization ability on more challenging tasks, achieving the highest scores on Olympiad-Bench and AIME24 across all models. This suggests that our meta objective encourages the model to leverage SFT only when it aids RL learning, ignoring supervised signals that lead to rote memorization and hinder generalization.

## 5. Analysis

**Training Dynamics.** We analyze the dynamics of mean reward and response length during training for RL, SFT→RL, BRIDGE, and SFT+RL on Qwen2.5-3B. As shown in Figure 3, the four methods exhibit markedly different patterns. RL suffers from online RL's sample inefficiency, showing slow growth in both response length and reward. Cold-start (SFT→RL) begins with extremely long responses due to SFT warm-up, *causing slow training* (see Table 4), followed by a sharp decline and gradual recovery. Despite starting with higher rewards, Cold-start's second-phase RL lacks guidance, resulting in final rewards similar to RL-Zero. SFT+RL mixes the two objectives directly without coordination, which slows RL training—both the mean reward and response length improve very slowly across the entire run. In contrast, BRIDGE benefits from continuous SFT guidance throughout training, enabling rapid reward growth that surpasses Cold-start and achieving superior convergence. These dynamics demonstrate that BRIDGE's bilevel optimization enables more efficient policy learning through sustained, targeted expert guidance.

**Training Efficiency.** We evaluate the cost-performance trade-offs by measuring wall-clock training time, average GPU memory usage per device, and final convergence performance across two model scales: Qwen2.5-3B (4×A100-

*Table 3.* Performance on Qwen3-8B-Base.

| Method | MATH 500 | Minerva Math | Olympiad Bench | AIME24 | AMC23 | Average |
|---|---|---|---|---|---|---|
| Base | 55.4 | 24.3 | 22.5 | 3.3 | 27.5 | 26.6 |
| SFT | 67.8 | 32.0 | 29.8 | 13.3 | 45.0 | 37.6 |
| RL | 76.2 | 36.0 | 42.4 | 10.0 | 50.0 | 42.9 |
| SFT→RL | 80.4 | 38.2 | 39.6 | 16.7 | 52.5 | 45.5 |
| SFT+RL | 72.2 | 34.2 | 39.2 | 10.0 | 45.0 | 40.1 |
| LUFFY | 75.4 | 36.4 | 43.1 | 10.0 | 55.0 | 44.0 |
| SRFT | 72.2 | 32.4 | 40.0 | 6.7 | 47.5 | 39.8 |
| CHORD | 76.6 | 37.5 | 42.2 | 13.3 | 60.0 | 45.9 |
| BRIDGE | 79.0 | 39.7 | 44.0 | 16.7 | 70.0 | 49.9 |

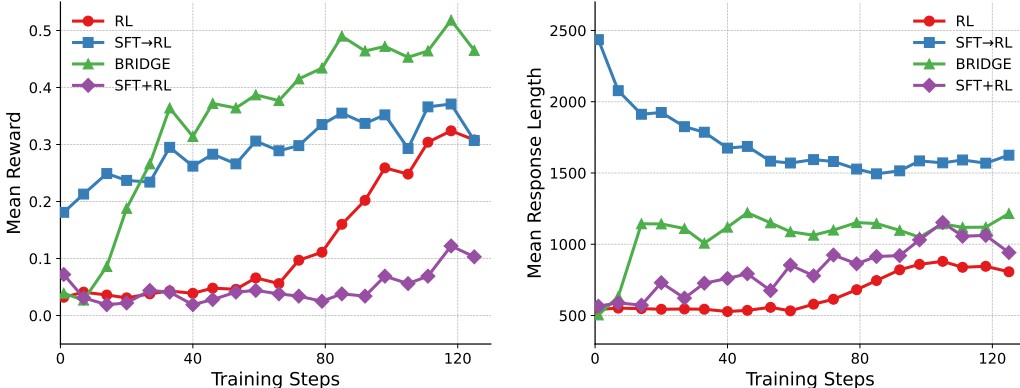

*Figure 3.* Training dynamics of mean reward and response length on Qwen2.5-3B.

*Table 4.* Cost-performance analysis on Qwen2.5-3B and Qwen3-8B-Base

| Metric | Qwen 2.5-3B | | | Qwen 3-8B-Base | | |
|---|---|---|---|---|---|---|
| | RL | SFT→RL | BRIDGE | RL | SFT→RL | BRIDGE |
| Time (hr) | 6.1 | 12.3 | 6.9 | 38.5 | 39.1 | 33.5 |
| Mem. (GB) | 52.2 | 45.9 | 59.3 | 50.7 | 60.8 | 67.4 |
| Acc. (%) | 32.2 | 32.2 | 36.0 | 42.9 | 45.5 | 49.9 |

80GB) and Qwen3-8B-Base (8×MI300-192GB). As shown in Table 4, two-stage pipeline (SFT→RL) requires nearly 2x the training time of RL, despite the short SFT stage. This overhead stems from long sequence lengths induced by the SFT stage (Fig. 3). BRIDGE achieved 44% and 14% time savings compared to the two-stage pipeline for the 3B and 8B models, respectively. Despite a modest 11% increase in memory usage for the larger model, BRIDGE consistently delivered superior performance improvements (11.8% for 3B and 9.7% for 8B models), demonstrating favorable cost-benefit trade-offs for practical deployment.

*Table 5.* Ablation of $\mathcal{J}_{\text{meta}}$ on Qwen3-8B-Base.

| Configuration | Average accuracy |
|---|---|
| BRIDGE | 49.9 |
| - w/o $\mathcal{J}_{\text{meta}}$ | 40.3 |

**Impact of the Meta-Objective ($\mathcal{J}_{\text{meta}}$).** We isolate the contribution of the meta-objective $\mathcal{J}_{\text{meta}}$ (Eq. 10) by disabling the upper-level update to assess the necessity of the bilevel formulation. As shown in Table 5, removing this term reduces BRIDGE to a naive multi-task learning between SFT and RL with fixed weighting, resulting in a substantial 9.6-point drop in average accuracy on Qwen3-8B-Base. This degradation confirms that simply combining objectives is insufficient; $\mathcal{J}_{\text{meta}}$ is critical for aligning supervised updates with the ultimate reinforcement learning goal, ensuring that SFT provides targeted assistance.

## 6. Generalization and Robustness

Our main experiments focus on mathematical reasoning. Because BRIDGE is *reward-agnostic*—requiring only a scalar

reward $R(\hat{y}, y)$ with no domain-specific assumptions—we now examine whether its benefits extend more broadly: to other reasoning domains and imperfect rewards scenarios.

**Generalization beyond mathematics.** To test transfer to non-mathematical reasoning, we train BRIDGE on Knights-and-Knaves logic puzzles, which require multi-step deduction under logical constraints, and hold out harder configurations (6–8 players) as an out-of-distribution (OOD) split. As shown in Table 6, BRIDGE beats RL at all difficulty levels (55.3 vs. 37.0 average), including a +12-point gain on the unseen OOD split, confirming that its cooperative formulation is not specific to mathematics.

*Table 6.* Generalization to logical reasoning (K&K) on Qwen2.5-3B. OOD denotes harder unseen configurations (6–8 players).

| Method | Easy (2–3) | Hard (4–5) | OOD (6–8) | Avg. |
|---|---|---|---|---|
| Base | 19.5 | 6.5 | 0.7 | 8.9 |
| RL | 53.0 | 40.0 | 18.0 | 37.0 |
| BRIDGE | 75.0 | 61.0 | 30.0 | 55.3 |

**Out-of-distribution generalization to code and science.** We further evaluate our trained checkpoints *without any additional training* on two non-math OOD benchmarks: Live-CodeBench (Jain et al., 2024) (code generation) and GPQA Diamond (Rein et al., 2024) (graduate-level science QA). As shown in Table 7, BRIDGE achieves the best average and is the only method that improves over the base model on both benchmarks. In contrast, both SFT and SFT→RL fall well below the base model, indicating that math-specific supervised training can damage general reasoning ability; BRIDGE avoids this by selectively incorporating only beneficial supervision through the meta-objective.

*Table 7.* Out-of-distribution generalization (pass@1) of Qwen2.5-3B checkpoints, evaluated with no additional training. LCB: Live-CodeBench; GPQA: GPQA Diamond.

| Method | LCB | GPQA | Avg. |
|---|---|---|---|
| Base | 32.95 | 32.32 | 32.64 |
| SFT | 20.74 | 21.72 | 21.23 |
| RL | 32.50 | 38.38 | 35.44 |
| SFT→RL | 23.86 | 25.76 | 24.81 |
| BRIDGE | 34.55 | 42.93 | 38.74 |

**Solution diversity.** A common concern is that RL's reward maximization collapses the policy onto a narrow set of solutions. We assess this via Pass@$K$ on two challenging benchmarks (Table 8): BRIDGE instead surpasses RL at every $K$, and the gap *widens* at larger $K$ (e.g., +10 points at Pass@32 on AIME24). BRIDGE therefore does not compress the solution space; instead, expert traces enrich exploration, helping it discover a broader and higher-reward set of solutions than pure RL reaches.

**Robustness to reward noise.** We simulate an imperfect

*Table 8.* Solution diversity via Pass@$K$ on Qwen2.5-3B. BRIDGE outperforms RL at every $K$, with the gap widening at larger $K$.

| Benchmark | Method | @4 | @8 | @16 | @32 |
|---|---|---|---|---|---|
| AIME24 | RL | 11.9 | 15.7 | 18.9 | 20.0 |
| | BRIDGE | 13.1 | 19.0 | 25.4 | 30.0 |
| OlympiadBench | RL | 40.4 | 47.3 | 53.5 | 59.0 |
| | BRIDGE | 42.2 | 49.6 | 56.1 | 61.6 |

verifier by independently flipping each binary reward with probability $p$ (Table 9). BRIDGE's advantage over RL *grows* with noise, from +3.8 at $p=0$ to +19.0 at $p=0.2$. This robustness is structural. Writing $J_{\mathrm{RL}}$ for the expected-reward objective, a binary reward flipped with probability $p$ has $\mathbb{E}[r_{\mathrm{noisy}}] = (1 - 2p)\, r_{\mathrm{true}} + p$, so the additive bias $p$ cancels in the cooperative-gain term and the meta-objective (Eq. 10) becomes, in expectation,

$$\mathbb{E}\left[J_{\mathrm{meta}}^{\mathrm{noisy}}\right] = \underbrace{(1 - \lambda)\, J_{\mathrm{SFT}}}_{\text{noise-independent}} + \lambda(1 - 2p) \underbrace{\left[J_{\mathrm{RL}}(\theta) - J_{\mathrm{RL}}(\hat{\theta})\right]}_{\text{cooperative gain}}.$$
(11)

Two effects make the signal robust. The supervised term is *entirely independent* of reward noise, since it learns from expert traces rather than rewards; and the cooperative-gain term retains its *sign*, with only its magnitude scaled by $1-2p$ (e.g., 60% survives at $p=0.2$, vanishing only at $p=0.5$, i.e., pure-random rewards). Hence BRIDGE's guidance remains reliable under imperfect verifiers.

*Table 9.* Robustness to reward noise on Qwen2.5-3B (five-math average); each binary reward flipped with probability $p$.

| Noise $p$ | BRIDGE | RL | Gap |
|---|---|---|---|
| 0% | 36.0 | 32.2 | +3.8 |
| 10% | 33.7 | 18.2 | +15.5 |
| 20% | 32.9 | 13.9 | +19.0 |

## 7. Related Work

**Post-training for reasoning.** Post-training for reasoning models typically follows two paradigms: supervised fine-tuning (SFT) and reinforcement learning (RL). While RL was first widely adopted to align models with human values (Chen et al., 2025c;b), recent reasoning models instead rely on reinforcement learning with verifiable rewards (RLVR), highlighting the critical role of RL in enhancing LLM reasoning (OpenAI; DeepSeek-AI et al., 2025; Chen et al., 2026). Recent studies analyze the trade-off between the two paradigms; for example, Chu et al. (2025) compare SFT and RL for reasoning tasks and find that RL generalizes significantly better, whereas SFT is prone to overfitting. A common practice therefore adopts a two-stage SFT→RL pipeline, where SFT is often used as a warm-up stage before RL. Within this recipe, the choice of supervised traces matters: SimpleRL (Zeng et al., 2025) observes that fine-tuning

on short-CoT datasets can harm reasoning ability, while He et al. (2025) find that long-CoT distilled data can improve the reasoning performance of smaller models when used as a warm-up stage before RL training. However, the advantage of the two-stage pipeline over pure RL is inconsistent, motivating tighter integration of the two learning signals.

**Integrating SFT and RL.** Recent efforts move beyond the decoupled "SFT then RL" recipe by mixing two objectives within one stage. Existing integration methods largely fall into two families. *Objective-level combination* mixes SFT and RL objectives via fixed or scheduled weights, including interleaved recipes (Ma et al., 2025) and single-stage hybrid training with adaptive reweighting or gating (Zhang et al., 2025b; Chen et al., 2025a; Fu et al., 2025). Representative examples include CHORD, which stabilizes training with global and token-wise reweighting (Zhang et al., 2025b), and SRFT, which mitigates interference via entropy-aware weighting and clipping (Fu et al., 2025). *Data-augmented RL* instead injects SFT demonstrations as off-policy guidance within RL (e.g., LUFFY) (Yan et al., 2025), often requiring additional mechanisms to handle distribution mismatch and data pairing constraints.

Despite empirical progress, most hybrids couple the signals heuristically and rarely characterize *when* a supervised update is actually beneficial for reward optimization. BRIDGE instead treats SFT–RL cooperation as a bilevel problem, meta-adapting the supervision to maximize the cooperative gain of joint training over RL alone and yields larger and more robust improvements than simple loss mixing. It offers a new perspective on integrating imitation and exploration for large reasoning models.

**Bilevel Optimization in LLMs.** Bilevel optimization (BLO) is a classical framework for modeling hierarchical learning problems, originating from Stackelberg leader-follower games. Two major classes of methods have been developed to solve BLO problems. Implicit gradient methods (Hong et al., 2020; Khanduri et al., 2021; Shen & Chen, 2022; Xiao et al., 2023) compute gradients through the lower-level problem using second-order derivatives. While theoretically robust, these methods are often computationally expensive and memory-prohibitive when applied to large-scale models such as LLMs. In contrast, penalty-based relaxation methods (Shen & Chen, 2023; Kwon et al., 2023; Shen et al., 2024; Lu, 2023) approximate the BLO formulation using only first-order gradients, making them substantially more scalable and thus better suited for LLM applications. Recent work has explored the use of bilevel optimization in LLMs for tasks such as data selection (Lin et al., 2024; Shen et al., 2025), inverse reinforcement learning (Li et al., 2024), and meta-learning (Choe et al., 2023; Shirkavand et al., 2025). To the best of our knowledge, our work is the first to cast reasoning-oriented LLM training as bilevel optimization, introducing a novel augmented model architecture for modeling and solving this problem. This provides a principled framework for integrating supervised and reinforcement learning, where SFT actively assists RL optimization rather than merely serving as warmup.

## 8. Conclusion

This work investigates how to effectively integrate supervised fine-tuning and reinforcement learning to improve the reasoning capabilities of LLMs. We begin by analyzing widely used training paradigms and identify a key limitation of existing baselines: two-stage pipelines decouple SFT from the reward-optimization phase, while naive single-stage objective mixing can be counterproductive because supervised updates are not uniformly beneficial for reward optimization. To address this, we introduce BRIDGE, a bilevel optimization framework that models SFT as the upper-level objective and RL as the lower-level objective. By employing a penalty-based relaxation, BRIDGE explicitly encourages joint training to outperform standalone RL, fostering tighter cooperation between the two learning paradigms. Empirical results on five mathematical reasoning benchmarks demonstrate that our method consistently outperforms strong baselines in both accuracy and training efficiency. Furthermore, extensive ablation studies confirm the necessity of the bilevel cooperation signal and the robustness of the method to hyperparameter variations. Overall, this work demonstrates that learning useful supervision is a viable and effective strategy for integrating SFT and RL, and that bilevel optimization offers an effective foundation for advancing reasoning-centric post-training methods.

## Limitations

*Evaluation scope:* while in principle BRIDGE only requires a scalar reward $R(\hat{y}, y)$, we target tasks with automatically verifiable answers (math, logic, code/science) and do not study open-ended generation with subjective rewards. *Scalability:* beyond our 3B–8B study, at larger scales (70B+) the auxiliary iterate $\hat{\theta}$ dominates memory while the LoRA teacher is negligible, and standard ZeRO-3 sharding with gradient checkpointing should keep BRIDGE practical.

## Impact Statement

This paper presents work intended to advance the field of machine learning, particularly methods for integrating supervised fine-tuning with reinforcement learning with verifiable rewards in reasoning models. BRIDGE meta-learns when expert-trace supervision is useful for reward optimization, improving training efficiency and stability as well as cost–performance trade-offs on mathematical reasoning

benchmarks. We do not anticipate risks unique to BRIDGE beyond those generally associated with improving the capabilities and accessibility of reasoning models, and we view continued community dialogue on responsible deployment of such systems as important.

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

# A. Additional Experiments

## A.1. Periodic SFT-Refresh Baseline

A natural question is whether the benefits of BRIDGE can be recovered by simply re-injecting supervision into the two-stage pipeline. We test this with a *periodic SFT-refresh* baseline: after the SFT warm-up, we periodically interleave SFT updates during the RL stage (every 5 or 10 steps). Table 10 shows that periodic refreshing does *not* improve over the standard SFT→RL pipeline and remains well below BRIDGE. Naively alternating SFT and RL does not provide the targeted, reward-aware supervision that BRIDGE's meta-objective learns; the gains therefore come from *learning which* supervised signals help RL, not merely from supplying more supervision.

*Table 10.* Periodic SFT-refresh baseline on Qwen2.5-3B (pass@1). Periodically injecting SFT updates during the RL stage does not match BRIDGE.

| Method | MATH500 | Minerva | Olympiad | AIME24 | AMC23 | Avg. |
|---|---|---|---|---|---|---|
| SFT→RL | 66.0 | 24.3 | 26.8 | 9.0 | 35.0 | 32.2 |
| + periodic SFT (freq=10) | 64.2 | 25.0 | 27.1 | 3.3 | 37.5 | 31.4 |
| + periodic SFT (freq=5) | 64.6 | 24.3 | 27.0 | 6.7 | 35.0 | 31.5 |
| BRIDGE | 66.2 | 23.9 | 28.9 | 13.3 | 47.5 | 36.0 |

## A.2. LoRA Configuration Robustness

To verify that the gains of BRIDGE are not artifacts of a specific parameter-efficient fine-tuning configuration, we ablate the LoRA rank $r$ and scaling factor $\alpha$ on Qwen3-8B-Base while keeping all other training settings fixed. Table 11 shows negligible variance across $(r, \alpha)$ choices, indicating that BRIDGE is robust to low-rank hyperparameter settings and that the improvements stem from the objective function itself.

*Table 11.* LoRA sensitivity ablation on Qwen3-8B-Base.

| $r/\alpha$ | MATH500 | Minerva | Olym. | AIME24 | AMC23 | **Avg.** |
|---|---|---|---|---|---|---|
| 32/16 | 79.0 | 39.7 | 44.0 | 16.7 | 70.0 | 49.9 |
| 16/32 | 79.0 | 38.6 | 44.0 | 16.0 | 70.0 | 49.5 |

## A.3. Sensitivity to the Mixing Weight $\lambda$

We analyze the weighting parameter $\lambda$, which controls the interpolation between the SFT and RL objectives in our penalty formulation (Eq. 6); we sweep $\lambda \in \{0.0, 0.3, 0.5, 0.7, 1.0\}$ on Qwen3-8B-Base (Table 12). At $\lambda = 0$ the formulation reduces to pure SFT and at $\lambda = 1$ to pure RL; these are *degenerate* endpoints rather than evidence of instability. Within the meaningful range $[0.3, 0.7]$, performance is consistently strong (47.2–49.9) and peaks at $\lambda = 0.5$, indicating that BRIDGE is robust to $\lambda$. The student learning rate $\alpha$ matches the baselines for fair comparison and the teacher learning rate $\beta$ follows the standard LoRA recommendation ($5 \times 10^{-5}$) without tuning, so $\lambda$ is the only BRIDGE-specific hyperparameter requiring adjustment.

*Table 12.* Sensitivity to $\lambda$ on Qwen3-8B-Base.

| $\lambda$ | 0.0 | 0.3 | 0.5 | 0.7 | 1.0 |
|---|---|---|---|---|---|
| Avg. | 37.6 | 47.2 | 49.9 | 49.0 | 42.9 |

