# OpenReview forum: "Beyond Two-Stage Training: Cooperative SFT and RL for LLM Reasoning"
_ICML.cc/2026/Conference — ICML 2026 regular_

### Official Review · Reviewer_2pFt · 2026-03-07

**Soundness:** 2
**Presentation:** 2
**Significance:** 3
**Originality:** 3
**Overall Recommendation:** 4
**Confidence:** 3

**Summary:**

The paper addresses the challenge of integrating supervised fine-tuning (SFT) and reinforcement learning with verifiable rewards (RLVR) to improve the reasoning capabilities of large language models. The authors propose BRIDGE, a bilevel meta-learning framework that introduces a lightweight LoRA module as a "teacher" whose meta-parameters are optimized to maximize a cooperative gain signal—the reward improvement of joint SFT–RL training over an RL-only baseline. Meanwhile, the base model "student" is updated via a convex combination of SFT and RL gradients. A penalty-based first-order relaxation avoids the need for expensive second-order computations. Empirically, BRIDGE consistently outperforms standard two-stage pipelines and recent single-stage hybrid methods (CHORD, SRFT, LUFFY) across three LLMs (Qwen2.5-3B, Llama-3.2-3B-Instruct, Qwen3-8B-Base) on five mathematical reasoning benchmarks, with gains of 2.8–4.0 average points over the strongest baselines.

**Compliance With Llm Reviewing Policy:**

Affirmed.

**Final Justification:**

During the rebuttal, the author provided a detailed discussion and experiments, which solved my concerns. I raise my score from 3 to 4, considering the contribution of the paper.

**Key Questions For Authors:**

Q1: The cooperative gain signal (Eq. 10) compares the joint model (θ) against an RL-only auxiliary model (θ-hat). Since θ-hat is updated with only RL gradients starting from the same initialization, the two models may diverge significantly over training. How sensitive is BRIDGE to this divergence, and have you considered periodically re-synchronizing θ and θ-hat?

Q2: The evaluation is restricted to mathematical reasoning with deterministic verifiers. How do you expect BRIDGE to perform in domains where reward signals are noisier or less well-defined (e.g., open-ended code generation, creative writing)? Does the cooperative gain signal remain meaningful when reward noise is high?

Q3: The paper focuses on models up to 8B parameters with an 11% memory overhead (over SFT→RL). What are the projected memory and compute requirements for scaling BRIDGE to 70B+ models? Is the auxiliary θ-hat model the primary bottleneck, and could strategies like gradient checkpointing or parameter sharing mitigate this?

Q4: Table 4 shows that BRIDGE is faster in wall-clock time than SFT→RL despite the additional per-iteration cost. Could you provide a more fine-grained analysis (e.g., tokens processed per second, GPU utilization) to help disentangle the effect of shorter response lengths from the inherent computational overhead of the bilevel updates?

**Limitations:**

Although the authors include limitations pertaining to the evaluation scope based on purely mathematical means in their conclusion, they do not specifically acknowledge the challenges related to scalability of larger models, the dependence upon deterministic verifiable reward structures, nor the memory overloads that would occur as part of some of those tasks. Additionally, including discussions regarding the issues noted above would enhance the impact of this paper. Finally, the authors provided an adequate description of the impact of the research on society within the Impact Statement section.

**Strengths And Weaknesses:**

**Strengths:**

1. The good narrative makes it easy for readers to follow through all the parts of the paper as well as understanding everything that is happening within the paper. The way that Figure 1 was created makes it very clear when comparing and contrasting how the proposed meta-learning process is creating methods at a higher level than using a two-stage or one-stage hybrid system, thus enabling the reader to quickly understand the teacher-student relationship being created in figure 1 through the manner in which they relate to each other.
2. The technical aspects and connection of the method to the use of bilevel optimization with large models is very solid due to the application of a penalty-based method (in the example it was based on first-order methods) providing a computationally feasible way to implement a theoretically complex bilevel process through large-scale LLM learning; in addition to having a clear derivation and solid connection to Danskin's theorem.
3. The data collected in the ablation study is a thorough study: the results of the J_meta ablation (Table 5) demonstrate that the proposed bilevel design is necessary (the average performance decreased significantly by 9.6 points from the average performance without J_meta), the λ sweep (Table 6) confirms that there is balanced coupling between both parts, and the rank/alpha analysis (Table 7) demonstrates that the hyperparameter values do not affect performance significantly.

**Weaknesses:**

1. There seems to be a sudden shift between the exact bilevel formulation (Eq. 4) and the penalty-based relaxation (Eqs. 5-6). Although there are mathematical definitions presented, it would be helpful for readers to have a more intuitive understanding of what the characteristics of the sub-optimality gap are that make it a good penalty term and how tight the relaxation will be under certain assumptions in order to support a wider audience's access to this paper.

2. Even though the title implies that there are significant enhancements to "Reasoning Models," the experimental results from coding only apply to mathematical evaluation metrics. Therefore, since no evaluation was done on anything related to coding activities (e.g. generating code, debugging code), or broader logical reasoning (e.g. providing commonsense reasoning, planning) it is difficult to support claims of universal effectiveness across a range of different types of reasoning.

3. The aforementioned approach results in a significant increase in memory utilization (11% versus SFT→RL for 8B model; ~33% compared to RL-only) as the memory requirement for both auxiliary parameter θ-hat and LoRA teacher parameter w must be kept in addition to the memory requirement for the base model (as illustrated in Table 4). This method is not adequately described for scaling to much larger models (for example, 700B parameters), where there are increasingly large memory constraints on running these methods.

4. The authors argue strongly against the use of a two-step pipeline because it loses the supervised signal after warm-up, resulting in the RL "reverting to unguided exploration" (Section 2.2). However, the authors do not show data from baseline experiments that have used a two-step pipeline plus simple techniques to potentially mitigate this issue (i.e. performing periodic SFT refreshes or replaying SFT data during the RL stage). This will allow for better isolation of whether BRIDGE's advantage is due to the meta-learning formulation or the provision of ongoing supervised signals throughout the entire training process as was done with the prior experiments.

5. The proposed system has a major reliance on "Reinforcement Learning with Verifiable Rewards," which here has been developed using deterministic rule-based matching techniques. However, there is little or no discussion of how the bilevel optimization structure performs in circumstances where reward signals are less readily available (for example, in open-ended or noisy situations), raising concerns about the stability and robustness of the proposed approach when the verifiability cannot be provided.

6. Although there is an emphasis on learning efficiency, the requirement for dual-batch processing results in approximately 50% greater computation for each forward/backward iteration than for typical single-objective hybrid approaches (3 primary computation vs. 2, per Algorithm 1). A more comprehensive evaluation of throughput (tokens per second) vs. total convergence time, will give a more complete picture of the actual computational trade-offs beyond merely the wall clock duration used for Table 4.

---

> ### Author Rebuttal · Authors · 2026-03-31
>
> We thank the reviewer for the thorough and thoughtful evaluation. We address each point below.
>
> **W1: Intuition for penalty relaxation.**
>
> The bilevel problem (Eq. 4) requires θ to be RL-optimal — intractable at LLM scale. The penalty relaxation (Eqs. 5-6) softens this: instead of demanding "θ must be the RL optimum," we penalize the sub-optimality gap p(w,θ) = max_{θ'} J_RL(θ',w) − J_RL(θ,w). Using p as a penalty is effective since:
>
> _Key properties:_ (a) p=0 iff θ is RL-optimal; (b) if p>0, ∇_θ p pushes θ toward RL optimality — making it a natural penalty.
>
> _Tightness:_ under standard smoothness assumptions, the penalized solution satisfies ‖θ_λ − θ*‖ = O(1−λ), where θ* is the bilevel optimum. Larger λ yields a tighter approximation at the cost of less SFT influence.
>
>
> **W2: Non-math evaluation.**
>
> We conducted experiments on additional reasoning domains:
>
> _(1) Logical reasoning._ We trained BRIDGE on Knights-and-Knaves logic puzzles (Qwen2.5-3B):
>
> | Method | Easy (2-3) | Hard (4-5) | OOD (6-8) | Avg |
> | :--- | :--- | :--- | :--- | :--- |
> | Base | 19.5 | 6.5 | 0.7 | 8.9 |
> | RL | 53.0 | 40.0 | 18.0 | 37.0 |
> | BRIDGE | 75.0 | 61.0 | 30.0 | 55.3 |
>
> BRIDGE outperforms RL across all difficulty levels including the OOD split, demonstrating generalization to logical constraint reasoning.
>
> _(2) Code and science reasoning._ We evaluated our models on LiveCodeBench (code generation) and GPQA Diamond (science QA) to test OOD generalization:
>
> | Method | LCB | GPQA | Avg. |
> | :--- | :---: | :---: | :---: |
> | Base | 32.95 | 32.32 | 32.64 |
> | SFT | 20.74 | 21.72 | 21.23 |
> | RL | 32.50 | 38.38 | 35.44 |
> | SFT→RL | 23.86 | 25.76 | 24.81 |
> | BRIDGE | 34.55 | 42.93 | 38.74 |
>
>  These demonstrate BRIDGE's effectiveness across different reasoning tasks.
>
>
>
> **W3/Q3: Memory and scalability to 70B+.**
>
> Our current experiments cover 3B and 8B models. For larger scales like 70B+, θ̂ is the primary memory bottleneck; the LoRA is negligible. We have not run 70B experiments, but we estimate that sharding and checkpointing can mitigate the overhead:
>
> (1) _ZeRO-3 sharding_: 70B RL training typically requires 64+ GPUs. With ZeRO-3, θ̂ (~140GB in bf16) is distributed across all devices, adding ~2GB per GPU — a manageable overhead. (2) _Gradient checkpointing_: more aggressive checkpointing on θ̂ can be applied to further reduce its memory footprint.
>
> We will add this discussion.
>
>
> **W4: Periodic SFT refresh baseline.**
>
> Thank you — we implemented this experiment. After SFT warmup, we periodically injected SFT updates during the RL stage:
>
> | Method | MATH500 | Minerva | Olympiad | AIME24 | AMC23 | Avg |
> | :--- | :--- | :--- | :--- | :--- | :--- | :--- |
> | SFT →RL | 66.0 | 24.3 | 26.8 | 9.0 | 35.0 | 32.2 |
> | + periodic SFT (freq=10) | 64.2 | 25.0 | 27.1 | 3.3 | 37.5 | 31.4 |
> | + periodic SFT (freq=5) | 64.6 | 24.3 | 27.0 | 6.7 | 35.0 | 31.5 |
> | BRIDGE | 66.2 | 23.9 | 28.9 | 13.3 | 47.5 | 36.0 |
>
> Periodic SFT refreshing does not improve over the two-stage method — naively alternating SFT and RL does not benefit the second-stage optimization. This highlights the necessity of meta-learning that learns which SFT signals help RL.
>
>
> **W5/Q2: Robustness to noisy rewards.**
>
> We tested this by injecting reward noise (flipping rewards with probability p) during training:
>
> | Noise | BRIDGE | RL | Gap |
> | :--- | :---: | :---: | :---: |
> | 0% | 36.0 | 32.2 | +3.8 |
> | 10% | 33.7 | 18.2 | +15.5 |
> | 20% | 32.9 | 13.9 | +19.0 |
>
> BRIDGE's advantage grows under noise (+3.8 → +19.0), as noise partially cancels in the reward-gap difference. This confirms the cooperative gain signal remains effective under imperfect rewards.
>
> **W6/Q4: Computational efficiency.**
>
> We report the fine-grained metrics requested:
>
> | Method | GPU Util (%) | Throughput (tok/s) |
> | :--- | :--- | :--- |
> | RL | 84.3 | 355.4 |
> | SFT → RL | 77.6 | 511.7 |
> | BRIDGE | 89.2 | 648.4 |
>
> While BRIDGE performs more computation per iteration, its throughput is actually the highest. This is because BRIDGE maintains moderate sequence lengths throughout training (Fig. 3), enabling more efficient GPU batching. In contrast, RL's short early-stage sequences underutilize the GPU, and SFT→RL's long SFT-induced sequences slow down rollout generation. Combined with the wall-clock results in Tab. 4 (6.9 vs. 12.3h), BRIDGE achieves the best cost-performance trade-off.
>
>
> **Q1: θ/θ̂ divergence.**
>
> In practice, θ and θ̂ diverge slowly due to PPO clipping, KL regularization, and the small learning rate. To test sensitivity, we re-synchronized θ̂←θ every 30 steps:
>
> | Method | MATH500 | Minerva | Olympiad | AIME24 | AMC23 | Avg |
> | :--- | :--- | :--- | :--- | :--- | :--- | :--- |
> | BRIDGE | 66.2 | 23.9 | 28.9 | 13.3 | 47.5 | 36.0 |
> | + re-sync@30 | 67.6 | 25.0 | 28.4 | 13.3 | 45.0 | 35.9 |
>
> Performance is comparable, confirming that the independently maintained θ̂ is a reliable baseline approximation.
>
> We will add a Limitations section covering evaluation scope, scalability, and reward verifiability.

---

> > ### Author Rebuttal · Reviewer_2pFt · 2026-04-02
> >
> > Thank you for your detailed rebuttal. I have several follow-up questions:
> >
> > 1. Regarding the experiment labeled “W5/Q2: Robustness to noisy rewards,” could the authors please clarify the exact value of p (i.e., the noise probability or level and how severe the noise is) and specify which RL method was used? Since BRIDGE demonstrates a remarkably large improvement in this setting, I would like to understand the details of the experiment in greater depth. Additional discussion and analysis of these results would be highly beneficial.
> >
> > 2. As a minor suggestion, I recommend providing a clearer explanation of why a LoRA module is chosen to serve as the “teacher” in the bilevel optimization framework for RL training. While LoRA is widely understood as a lightweight, parameter-efficient fine-tuning method, its role here as a meta-parameter that meta-learns targeted supervision feels like a conceptual gap for some readers. Elaborating on this design choice (e.g., its separation from the student LLM parameters and its specific function in maximizing the reward-gap signal) would make that section more intuitive and accessible.

---

> > > ### Author Response · Authors · 2026-04-03
> > >
> > > **Experimental protocol.** We inject reward noise by independently flipping each sample's binary reward with probability p: correct (1) → incorrect (0) and vice versa. This simulates a noisy verifier. The base RL algorithm is GRPO (consistent with the main experiments). We evaluate on the same 5 math benchmarks. We tested p=0.1 (10%) and p=0.2 (20%).
> > >
> > > We analyze the impact of reward noise on both levels of BRIDGE:
> > >
> > > _Upper-level (cooperative gain)._ For binary rewards with flip probability p:
> > > ```
> > > E[r_noisy] = r_true·(1-p) + (1-r_true)·p = r_true·(1-2p) + p
> > > ```
> > > Therefore:
> > >
> > > ```
> > > E[J_meta_noisy] = E[J_RL_noisy(θ)] - E[J_RL_noisy(θ̂)]
> > >                = [(1-2p)·J_true(θ) + p] - [(1-2p)·J_true(θ̂) + p]
> > >                = (1-2p) · J_meta_true
> > > ```
> > > The bias term p cancels in the difference. The signal direction is preserved — only its magnitude is attenuated by (1−2p). At p=0.2, 60% of the signal is retained; only at p=0.5 (pure random) does it vanish. This ensures the meta-learning signal remains reliable under noise.
> > >
> > > _Lower-level (gradient fusion)._ While the RL gradient is corrupted by noise, the SFT component provides reward-independent supervision from expert traces, offering a stable learning signal regardless of reward quality.
> > >
> > > Together, these explain why BRIDGE degrades gracefully under noise reward settings.
> > >
> > > **LoRA as teacher.** Thank you for the suggestion. The key design rationale is gradient isolation: by placing meta-parameters in a separate LoRA module, J_meta updates only w without interfering with θ's RL optimization. Since w is part of π_{θ+w}, changing w alters the loss landscape that θ optimizes over, effectively shaping which SFT gradients are beneficial for RL. At inference time, w is merged into θ with zero additional overhead. Table 7 confirms low-rank capacity suffices for this role (rank 16 ≈ rank 32). We will expand this discussion in the revision.
> > >
> > > We sincerely appreciate your constructive feedback, which has greatly strengthened our work. We will incorporate all suggestions in the revision. We welcome any further questions. Thank you again for your review.

---

### Official Review · Reviewer_h4wc · 2026-03-12

**Soundness:** 3
**Presentation:** 2
**Significance:** 2
**Originality:** 3
**Overall Recommendation:** 4
**Confidence:** 4

**Summary:**

The paper introduces BRIDGE, a scalable meta-learning framework designed to seamlessly integrate Supervised Fine-Tuning (SFT) and Reinforcement Learning with Verifiable Rewards (RLVR) for training Large Reasoning Models (LRMs). Recognizing that naive combinations or two-stage pipelines (SFT → RL) can lead to suboptimal reward maximization, the authors formulate the integration as a bi-level Stackelberg game. In this setup, SFT acts as an upper-level teacher (updating a lightweight LoRA module) that learns to maximize the "reward-gap"—the improvement of the joint SFT-RL objective over a pure RL baseline. Meanwhile, the lower-level student (the base LLM) updates its parameters using a fused gradient. To make this computationally feasible at the LLM scale, the authors employ a first-order penalty-based relaxation. Evaluated on Qwen2.5-3B, Llama-3.2-3B, and Qwen3-8B across diverse math benchmarks, BRIDGE demonstrates consistent improvements in both accuracy and training efficiency compared to existing baselines.

**Compliance With Llm Reviewing Policy:**

Affirmed.

**Final Justification:**

The rebuttal addressed my main concerns. I raise my score to 4

**Key Questions For Authors:**

# Questions

### 1. Table 3 Verification:

Please check the raw data and correct the metric order to ensure reliable baseline comparisons.

### 2. Response Diversity Metrics:

It is recommended to include metrics such as Pass@K vs. Pass@1 gap, average sequence diversity over N rollouts, or entropy of reasoning paths to assess whether BRIDGE mitigates or exacerbates RL mode collapse.

### 3. Hyperparameter Tuning Guidelines:

Providing practical heuristics or guidance for tuning α, β, and λ would improve the method’s usability across different tasks or models.

### 4. Domain Generalization: Beyond Math Tasks

All experiments are conducted in the mathematical reasoning domain. Could the authors comment on how BRIDGE would generalize to other reasoning tasks (e.g., coding, or general-domain reasoning)? Are any modifications to the SFT teacher or RL reward design needed for other domains?

**Limitations:**

Yes

**Strengths And Weaknesses:**

# Strengths
Strengths

### 1. Novel and Principled Modeling Approach

The shift from heuristic objective weighting methods (e.g., CHORD, SRFT) to a bi-level optimization framework represents a highly principled approach to resolving the objective conflict between SFT and RL.

This design is theoretically more rigorous and facilitates a systematic treatment of interference issues in multi-objective optimization.

### 2. Computational Efficiency

By adopting a penalty-based relaxation method and restricting the upper-level teacher to LoRA modules, the authors successfully avoid the computationally expensive second-order derivative calculations typically required in bi-level optimization.

This design significantly reduces computational complexity while maintaining the method’s expressive power, making the algorithm more scalable.

### 3. Robust Experimental Results

Key experimental results demonstrate that:
    •    BRIDGE consistently outperforms the following across different model families:
    •    Traditional two-stage training workflows
    •    State-of-the-art single-stage hybrid methods
    •    Additionally, compared to the standard SFT→RL workflow, BRIDGE significantly reduces actual training time (wall-clock time).

This indicates that the method achieves a good balance between performance and efficiency.

### 4. In-Depth Dynamic Analysis

An analysis of training dynamics (Figure 3) clearly explains why BRIDGE accelerates training:
    •    It avoids the sequence-length bloat commonly observed in SFT during the cold-start phase
    •    This makes RL rollouts more efficient
    •    Improving overall training efficiency

This analysis enhances the method’s interpretability.

# Weaknesses
### 1. Reporting Error in Table 3

In Table 3 (Qwen3-8B-Base Performance):

The “SFT” baseline row reports:
    •    AIME24 score of 45.0
    •    AMC23 score of 13.3

These results are inconsistent with the known relative difficulty levels of these competitions.

This likely indicates:
    •    The metrics have been swapped
    •    Or there is an error in the data recording

This issue may undermine the reliability of the baseline comparison.

### 2. Risk of Reduced Response Diversity (Mode Collapse)

Pure RLVR typically emphasizes “exploitation,” which often leads to:
    •    Mode collapse
    •    Policies converging to an extremely narrow distribution of inference paths
    •    Reduced generative diversity

In this two-layer framework, because:
    •    The SFT teacher is explicitly updated to maximize the RL reward gap

There is a risk that:
    •    the teacher module may also collapse
    •    thereby losing the original diverse inference trajectories present in the SFT data

This may affect the model’s generalization ability and diversity.

### 3. Hypersensitivity

Table 6 shows:

The mixing coefficient λ has a significant impact on performance, for example:
    •    λ = 0.5 → 49.9
	•    λ = 0.0 → 37.6

Additionally, the algorithm depends on:
    •    α (student learning rate)
    •    β (teacher learning rate)
    •    λ (mixing coefficient)

Therefore, BRIDGE may be sensitive to initialization and hyperparameters in new domains, requiring careful tuning.

---

> ### Author Rebuttal · Authors · 2026-03-31
>
> We thank the reviewer for the detailed evaluation. We address each point below.
>
>
> **W1/Q1: Table 3 error.**
>
>
> Thank you for catching this. Confirmed — the two values in the SFT baseline were swapped. The average and conclusions are unaffected.
>
>
> **W2/Q2: Response diversity.**
>
> We computed Pass@K of Qwen2.5-3B on two challenging benchmarks:
>
> Table R1. Pass@K diversity analysis
> | Benchmark | Method | Pass@4 | Pass@8 | Pass@16 | Pass@32 |
> | :--- | :--- | :--- | :--- | :--- | :--- |
> | AIME24 | RL | 11.9 | 15.7 | 18.9 | 20.0 |
> |  | BRIDGE | 13.1 | 19.0 | 25.4 | 30.0 |
> | OlympiadBench | RL | 40.4 | 47.3 | 53.5 | 59.0 |
> |  | BRIDGE | 42.2 | 49.6 | 56.1 | 61.6 |
>
> BRIDGE outperforms RL at every K, with the gap widening at higher K. This indicates our method does not compress the solution space. Instead, _expert reasoning traces from SFT data effectively enrich the policy's exploration space, enabling it to discover a broader and higher-reward set of solutions that pure RL struggles to reach_.
>
> Notably, our OOD generalization evaluation (see Table R4 in W4) further supports this — BRIDGE generalizes better than baselines on unseen domains.
>
>
>
> **W3/Q3: Hyperparameter sensitivity.**
>
> We would like to clarify that λ=0 and λ=1 in Table 6 represent **_degenerate cases_**, not sensitivity: λ=0 reduces BRIDGE to pure SFT, and λ=1 reduces to pure RL. Within the meaningful range [0.3, 0.7], performance is consistently strong (47.2–49.9), showing that BRIDGE is robust to λ.
>
> For the remaining hyperparameters: **the student/LLM LR (α) is set identical to all baselines for fair comparison**, and the teacher LR (β) follows the standard LoRA recommendation (5e-5) without tuning. **In practice, λ is the only hyperparameter requiring tuning**, for which Table 6 provides detailed guidance.
>
> To further demonstrate this, **we applied the exact same hyperparameters across all new domains in W4** (Tables R2–R4) and achieved consistently strong results.
>
> **W4/Q4: Generalization beyond math.**
>
> BRIDGE is designed to be reward-agnostic — it only requires a scalar reward R(ŷ,y) with no domain-specific assumptions. We conducted three additional experiments, _all using the same hyperparameters as our math experiments_:
>
> _**(1) New domain: logical reasoning**._ We applied BRIDGE to Knights-and-Knaves logic puzzles on Qwen2.5-3B, with an OOD split on harder unseen configurations (6–8 players):
>
> Table R2. Results on Knights-and-Knaves
>
> | Method | Easy (2-3) | Hard (4-5) | OOD (6-8) | Avg |
> | :--- | :--- | :--- | :--- | :--- |
> | Base | 19.5 | 6.5 | 0.7 | 8.9 |
> | RL | 53.0 | 40.0 | 18.0 | 37.0 |
> | BRIDGE | 75.0 | 61.0 | 30.0 | 55.3 |
>
> BRIDGE outperforms RL across all difficulty levels including the OOD split, demonstrating generalization to logical reasoning.
>
> _**(2) Reward noise robustness**._ To test whether BRIDGE extends to domains with imperfect reward signals, we injected reward noise (flipping rewards with probability p) during training on Qwen2.5-3B:
>
> Table R3. Robustness to reward noise
>
> | Noise | BRIDGE | RL | Gap |
> | :--- | :---: | :---: | :---: |
> | 0% | 36.0 | 32.2 | +3.8 |
> | 10% | 33.7 | 18.2 | +15.5 |
> | 20% | 32.9 | 13.9 | +19.0 |
>
> BRIDGE's advantage over RL grows under noise (+3.8 → +19.0), confirming robustness to imperfect reward signals.
>
> _**(3) Non-math OOD generalization**._ We evaluated our checkpoints on *LiveCodeBench (code)* and *GPQA Diamond (science)* without additional training:
>
> Table R4. Results on non-math OOD benchmarks
>
> | Method | LCB | GPQA | Avg. |
> | :--- | :---: | :---: | :---: |
> | Base | 32.95 | 32.32 | 32.64 |
> | SFT | 20.74 | 21.72 | 21.23 |
> | RL | 32.50 | 38.38 | 35.44 |
> | SFT→RL | 23.86 | 25.76 | 24.81 |
> | BRIDGE | 34.55 | 42.93 | 38.74 |
>
>
> These results confirm that no modifications to the teacher or reward design are needed — it works out-of-the-box given a scalar reward R(ŷ,y), as demonstrated across different tasks and evaluation settings, with good robustness to varying reward quality.

---

> > ### Author Rebuttal · Reviewer_h4wc · 2026-04-04
> >
> > Thanks for feedback, I will increase my score to 4.

---

### Official Review · Reviewer_M8d7 · 2026-03-13

**Soundness:** 4
**Presentation:** 3
**Significance:** 3
**Originality:** 3
**Overall Recommendation:** 5
**Confidence:** 3

**Summary:**

This paper introduces BRIDGE, a meta-learning framework designed to harmonize supervised fine-tuning and reinforcement learning with verifiable rewards. Unlike traditional two-stage or naive hybrid approaches, BRIDGE treats SFT as a selective "teacher" that guides the RL "student" only when SFT updates directly contribute to reward gains. This is implemented via a bilevel optimization objective where LoRA parameters adaptively modulate the supervision signal. Evaluated on five math reasoning benchmarks across multiple LLMs, the method demonstrates superior performance in terms of reward, accuracy, and training stability compared to existing single-stage and pipeline baselines.

**Compliance With Llm Reviewing Policy:**

Affirmed.

**Final Justification:**

Thank you for your rebuttal and I'd like to keep my score.

**Key Questions For Authors:**

1. When generating rollouts for computing RL or reward-gap, do you use both θ and w together to sample, or do you only use θ?
2.

**Limitations:**

Consider noting potential biases or overfitting to the reward metric or SFT traces, which could limit real-world applicability or induce undesirable behaviors.

**Strengths And Weaknesses:**

Strengths
- Well-written and clearly structured, making the algorithm and experiments easy to follow despite the complexity of bilevel optimization.
- Addresses a real problem: naive combination of SFT + RL can hurt RL optimization; BRIDGE provides a principled solution.
- Bilevel meta-learning is novel in this context.
- Shows consistent empirical improvements across multiple LLM families (Qwen, Llama) and model sizes (3B–8B).
- Includes training efficiency analysis, showing faster convergence and reduced wall-clock time compared to two-stage SFT→RL pipelines.

Weaknesses
- The bilevel update rules and θ vs. LoRA w updates are not intuitive; Figure 1 helps but more explicit pseudocode would improve readability.
- The rationale for using a LoRA module as the teacher is not fully justified. While the method is effective, it is unclear why a lightweight adapter is sufficient to capture which SFT updates are beneficial for RL optimization. More discussion or ablation on the necessity of LoRA as the meta-parameter would improve clarity.
- Some theoretical aspects are confusing. For example, in Eq. 9 the computation of the reward-gap term J_{RL}(\theta, w) - J_{RL}(\hat{\theta}, w) is described, but it is unclear whether the same rollout is used for both θ and θ̂ or if new trajectories are sampled. Clarifying how the reward-gap is computed and which trajectories are used would improve reproducibility.

Typo
- On page 4, line 218, the sentence ends with a comma but should end with a period.

---

> ### Author Rebuttal · Authors · 2026-03-31
>
> We sincerely thank the reviewer for the positive and thoughtful assessment.
>
> **W1: Pseudocode readability.**
>
> Thank you. We have prepared a detailed diagram illustrating the optimization flow (available at https://anonymous.4open.science/r/anon-DEFE/Detailed_diagram.png). We will also add explicit equation references to Algorithm 1 (e.g., annotating the θ update with Eq. 7 and the w update with Eq. 10) in the revision.
>
>
> **W2: LoRA as the teacher.**
>
> Since w is part of the model (π_{θ+w}), changing w alters the loss landscape that θ optimizes over, thereby shaping the SFT and RL gradients that θ receives. The meta-objective finds w values that make the SFT gradient more aligned with RL optimization. We also tested different LoRA capacities: Table 7 shows that rank 16 and 32 yield similar results, empirically confirming that low-rank capacity suffices for this role. Additionally, separating w from θ provides clean gradient isolation, preventing the meta-objective from interfering with θ's RL optimization.
>
>
> **W3/Q1: Rollout sampling in Eq. 9.**
>
> Both θ and θ̂ are evaluated on the same trajectories sampled from π_{θ+w}. We sample one batch per iteration and compute log-probabilities under both models on these shared trajectories. We will clarify this in the revision.
>
>
> **Typo (p4, l218):** Thank you — will fix.
>
>  **Limitation.** Thank you for the suggestion. We will add a discussion of potential biases in the revision.

---

> > ### Author Rebuttal · Reviewer_M8d7 · 2026-04-02
> >
> > Thank you for your rebuttal and I'd like to keep my score.

---

### Official Review · Reviewer_64ED · 2026-03-13

**Soundness:** 3
**Presentation:** 2
**Significance:** 3
**Originality:** 2
**Overall Recommendation:** 3
**Confidence:** 3

**Summary:**

This paper proposes BRIDGE, a framework that learns supervision useful for reinforcement learning with verifiable rewards. Instead of naively combining SFT and RL, it trains a lightweight teacher to provide reward-improving guidance. Across multiple math reasoning benchmarks and model scales, BRIDGE consistently outperforms pure RL, two-stage SFT→RL, and prior hybrid methods.

**Compliance With Llm Reviewing Policy:**

Affirmed.

**Final Justification:**

The rebuttal addressed my main concerns. I raise the Soundness score to 3 and Presentation score to 2.

**Key Questions For Authors:**

- How robust is the generalization capability of the BRIDGE method? Please include evaluations on some Out-of-Distribution tasks.
- What value was used for $\lambda$ in the main experiments? Please clarify this explicitly.
- Regarding Figure 2: What would the outcome be if training were continued for a few more steps? Would the RL approach eventually outperform the SFT-then-RL baseline?
- Is the teacher or the student model used for final inference? Please compare the performance differences between the two when applied to inference tasks.

**Limitations:**

Limitations are not discussed in the paper. The authors could primarily supplement the text with relevant explanations and experiments addressing the points raised in the "Questions" section.

**Strengths And Weaknesses:**

- In terms of Soundness, this work is generally solid and features extensive experimentation; however, it does exhibit a few weaknesses:
    - It lacks an evaluation of the proposed method's generalization capabilities.
- Regarding Presentation, this work suffers from several weaknesses:
    - The text and components within Figure 1 are extremely blurry.
    - A detailed schematic diagram of BRIDGE is missing.
    - The specific value chosen for the parameter $\lambda$ is not explicitly stated.
- In terms of both Significance and Originality, this work is adequate, presenting no particularly glaring weaknesses.

---

> ### Author Rebuttal · Authors · 2026-03-31
>
> We thank the reviewer for the constructive feedback. We address each concern below.
>
>
> **Q1: OOD generalization.**
>
> We include evaluations on two OOD benchmarks—LiveCodeBench (code generation) and GPQA (graduate-level science QA)—without any additional training.
>
> Table R1. Results of Qwen2.5-3B on OOD benchmarks
> | Method | LCB | GPQA | Avg. |
> |--------|-----|------|------|
> | Base | 32.95 | 32.32 | 32.64 |
> | SFT | 20.74 | 21.72 | 21.23 |
> | RL | 32.50 | 38.38 | 35.44 |
> | SFT→RL | 23.86 | 25.76 | 24.81 |
> | BRIDGE | 34.55 | 42.93 | 38.74 |
>
>
> BRIDGE outperforms all baselines on both OOD benchmarks. Notably, both SFT and SFT→RL fall well below the base model, indicating that math-specific supervised training damages general reasoning ability. BRIDGE avoids this by selectively incorporating only beneficial SFT signals through the meta-objective.
>
>
>
> **Q2: Value of λ.**
>
> We apologize for the omission. The main experiments use λ=0.5, as identified optimal in Table 6. We will state this explicitly in experimental details.
>
>
> **Q3: Would RL eventually outperform SFT→RL if training continued?**
>
> We extended training to 200 steps:
>
> | Step | 0 | 40 | 80 | 120 | 160 | 200 |
> |------|-----|-----|------|------|------|------|
> | RL | 0.032 | 0.039 | 0.111 | 0.308 | 0.313 | 0.313 |
> | SFT→RL | 0.181 | 0.262 | 0.335 | 0.307 | 0.320 | 0.316 |
>
> The two methods converge to similar levels by step 120, with SFT→RL's early advantage largely vanishing. This confirms that SFT warm-up helps initialization but provides diminishing returns — motivating sustained supervision beyond warm-up.
>
>
> **Q4: Teacher vs. student for inference.**
>
> We use the student (θ) with teacher LoRA (w) merged, following standard LoRA practice — this adds _zero inference overhead_ since LoRA weights are absorbed into the base model. The two components cannot be used independently: training rollouts are sampled from π_{θ+w}, so separating them at inference creates a train-test mismatch.
>
>
>
> **Presentation.**
>
> The blurriness in Figure 1 was caused by PDF compression; we will provide the high-resolution version and a detailed schematic figure in the revision (available at https://anonymous.4open.science/r/anon-DEFE/Detailed_diagram.png). We will also add a dedicated Limitations section.

---

> > ### Author Rebuttal · Reviewer_64ED · 2026-04-03
> >
> > Thank you to the author for the detailed response. All of my questions have been resolved. I will raise the Soundness and Presentation scores.

---

### Decision · Program_Chairs · 2026-04-30

**Decision:**

Accept (regular)

**Comment:**

The paper proposes BRIDGE, which frames the integration of SFT and RLVR as a bilevel meta-learning problem: a lightweight LoRA teacher is trained in the outer loop to optimize a meta objective, while the inner-loop model is updated with joint SFT and RL gradients. The method shows consistent gains over sequential first SFT then RL training and joint training baselines on math reasoning benchmarks.

Reviewers agreed on the significance and originality of the approach. The rebuttal addressed concerns about OOD generalization and mode collapse of the method through additional experiments, and demonstrated that a periodic SFT refresh baseline does not match BRIDGE’s performance. The bilevel optimization framework is both principled and novel in the post-training setting. We therefore recommend acceptance.